# A Review on Nonlocal Theories in Fatigue Assessment of Solids

**DOI:** 10.3390/ma16020831

**Published:** 2023-01-15

**Authors:** Saeed H. Moghtaderi, Alias Jedi, Ahmad Kamal Ariffin

**Affiliations:** 1Department of Mechanical and Manufacturing Engineering, Faculty of Engineering and Built Environment, Universiti Kebangsaan Malaysia, Bangi 43600, Selangor, Malaysia; 2Centre for Automotive Research, Faculty of Engineering and Built Environment, Universiti Kebangsaan Malaysia, Bangi 43600, Selangor, Malaysia

**Keywords:** nonlocal theories, continuum damage model, fatigue life and damage, fracture mechanics, material failure, solid structures

## Abstract

A review of nonlocal theories utilized in the fatigue and fracture modeling of solid structures is addressed in this paper. Numerous papers have been studied for this purpose, and various nonlocal theories such as the nonlocal continuum damage model, stress field intensity model, peridynamics model, elastic-plastic models, energy-based model, nonlocal multiscale model, microstructural sensitive model, nonlocal lattice particle model, nonlocal high cycle fatigue model, low cycle fatigue model, nonlocal and gradient fracture criteria, nonlocal coupled damage plasticity model and nonlocal fracture criterion have been reviewed and summarized in the case of fatigue and fracture of solid structures and materials.

## 1. Introduction

Fatigue assessment of solid structures and materials is of considerable importance since fatigue performance and behavior are strongly linked to microscopic issues like crack initiation and propagation. Fatigue is a concentrated and progressive process in which structural damage accumulates constantly due to external loadings such as cyclic loads, resulting in irreversible microplastic deformation, crack propagation, and final failure. Furthermore, reliable damage prediction is problematic owing to the large number of parameters, including environmental and boundary conditions, complex and multiaxial loading conditions, geometrical discontinuities, and micro-structural problems that lead to *size effect* phenomena [1,2,3,4,5,6,7].

The mechanical behavior of solid structures and materials, including fatigue and fracture modeling of micro/nano-structures, cannot be fully characterized via the classical theory of elasticity because it cannot describe the mechanical characteristics of micro/nano-scale materials. Hence, in microscopic analyses of field quantities, numerous size-dependent theories have been developed in the literature. These theories have been facilitated with appropriate intrinsic length scales, so they are capable of capturing size effect phenomena [8,9,10,11].

Size-dependent models can be classified into various frameworks [12,13] such as the nonlocal elasticity theory, which is intertwined with Eringen’s ground-breaking 1984 work [14] on evaluating the mechanical response of micro/nano-structures. This theory has been exploited for investigations of elastic waves lattice dispersion, wave propagation in nano-composites, dislocation mechanics, fracture mechanics, surface tension fluids, and other related topics. In this approach, to interpret the long-range nonlocal effects, an integral convolution between the strain field and a nonlocal kernel is applied [15]. However, due to conflict between the constitutive and equilibrium boundary conditions, the resulting differential equation of nonlocal problems associated with the nonlocal elasticity model led to ill-posed elastostatic problems.

The majority of classical continuum theories are founded on hyper-elastic constitutive relations, which indicate that the stress at a given point is a function of the strain at that point. In contrast, nonlocal continuum mechanics emphasizes that the stress at a point is a function of the strains at all points along the continuum. These theories incorporate information about the forces between atoms, and the internal length scale has been introduced into constitutive equations as a material parameter [16,17,18].

Continuum damage mechanics [19] has been used to characterize fatigue crack initiation and propagation in general. It provides a macroscopic field variable that reflects the average microstructural material damage [20,21]. This so-called “damage variable” appears in the constitutive relations that regulate a material’s deformation behavior [22]. At the crack initiation stage, the damage variable grows progressively under the impact of mechanical loading. After a certain amount of accumulated damage, the material can no longer transfer stresses locally and a crack is initiated. Damage then continues to grow around the crack and it can propagate through the process of stress redistribution and damage growth. However, continuum models depend on this approach to allow for discontinuity solutions, wherein damage develops locally on a surface while the surrounding material remains unaffected. This damage localization contradicts the assumed smoothness of the damage field, affecting the physical relevance of damage modeling.

The present study was conducted to investigate nonlocal theories already presented in the literature to ascertain fatigue and fracture within solid structures and materials. First, the nonlocal theories and their applications are introduced. The second section presents an overview of the nonlocal theories commonly used for fatigue and fracture purposes, namely the nonlocal continuum damage model, stress field intensity model, peridynamics model, elastic-plastic models, and nonlocal multiscale model. Then, the frameworks used more occasionally—such as the microstructural sensitive model, nonlocal lattice particle model, nonlocal high cycle fatigue model, optimized effective damage parameter, nonlocal and gradient fracture criteria, nonlocally coupled damage plasticity model, and nonlocal fracture criterion—are reviewed. Finally, the current study is summarized.

## 2. Nonlocal Theories in Fatigue Evaluation

In this section, the nonlocal theories and models used in fatigue evaluation of solid structures and materials that were deemed more practical and demanding in the literature are categorized and discussed.

### 2.1. Nonlocal Continuum Damage Model

The nonlocal continuum damage model (NCDM), introduced by Cabot and Bazant in 1987 [23], was also developed by the same authors [24] to expose nonlocal treatment to factors that affect the strain softening of the material. For this purpose, they substituted the usual local damage energy release rate for its special average over the representative volume of material, whose size was a characteristic of the material. From the thermodynamic perspective, the state of a material can be described by its free energy density ψ as follows:(1)$$\rho \psi =\frac{1-D}{2}\varepsilon :C:\varepsilon$$
where ρ, D, ε, C, and : are the mass density, damage tensor, strain tensor, stiffness tensor, and tensor product, respectively. Since damage comprises the creation and propagation of cracks and voids, the growth of damage dissipates energy. The energy dissipation rate φ is given as follows:(2)$$\varphi =-\frac{\partial \left(\rho \psi \right)}{\partial t}=-\frac{\partial \left(\rho \psi \right)}{\partial D}\;\frac{\partial D}{\partial t}=Y\dot{D}$$
in which the damage energy release rate Y, is given below.
(3)$$Y=-\frac{\partial \left(\rho \psi \right)}{\partial D}=\frac{1}{2}\varepsilon :C:\varepsilon$$

According to the second law of thermodynamics, φ≥0. This condition is always verified since Y is a positive definition function of ε and the damage rate $\dot{D}\geq 0$.

In particular, the dynamic failure caused by wave propagation in a material happens when the material is just beginning to soften, without any stable progressive accumulation of damage before the failure and without any dissipation energy. To circumvent this problem, the damage energy release rate is defined in a nonlocal manner.
(4)$$\bar{Y}=\frac{1}{V_{r}-V^{*}}\underset{V_{r}-V^{*}}{\int }YdV$$
where $\bar{Y}$ represents the mean of Y over the representative volume $V_{r}$ of the material centered around the given point. Additionally, because of spatial averaging, locations at or extremely close to the boundary must receive specific treatment. Therefore, $V^{*}$ is the part of the material that extends beyond the boundary as a representative volume.

In the case of a one-dimensional elastic bar 0≤x≤L with Young’s modulus E and a characteristic length ξ, the damage energy formulation and characteristic length limitations are reduced to
(5)Y¯(x)=1x2−x1∫x1x212Eε2(x)dxx1=0 → x2=x+ξ2x1=x−ξ2 → x2=L

In 1997, Jirasek [25] numerically analyzed this nonlocal model for a simple one-dimensional bar. The results revealed that averaging the damage energy release rate was less expensive from a computational perspective as it is a scalar quantity. Averaging the energy also appeared to be more rational from a physics standpoint.

Based on an alternative view of the nonlocal damage model developed by Cabot and Bazant [23] to consider the mean fracturing strain, Peerlings et al. [26] defined a nonlocal equivalent strain $\bar{\varepsilon }\left(x\right)$ as follows:(6)$$\bar{\varepsilon }\left(x\right)=\frac{1}{\Gamma \left(x\right)}\int_{V}W\left(d,\xi \right)\tilde{\varepsilon }\left(s\right)dV$$
where $\tilde{\varepsilon }$ is a positive equivalent measure of the strain state and the factor Γ(x) is defined as
(7)$$\Gamma \left(x\right)=\int_{V}W\left(d,\xi \right)dV$$
in which W(d,ξ) is a weight function defined as
(8)$$W\left(d,\xi \right)=\frac{1}{{\left(2\pi \right)}^{3/2}\xi^{3}}exp\left(-\frac{d^{2}}{2\xi^{2}}\right)$$
where d=|x−s| represents the distance between points x and s, which is shown in Figure 1 below as the schematic of a sample with a crack to describe the nonlocal region near a crack.

To obtain an effective differential form, the local equivalent strain $\tilde{\varepsilon }$ in Equation (6) can be substituted with a reduced Taylor expansion, followed by some mathematical manipulation, which leads to the aforementioned approximation of relation
(9)$$\bar{\varepsilon }-\frac{\xi^{2}}{2}{▽}^{2}\bar{\varepsilon }=\tilde{\varepsilon }$$
where ${▽}^{2}=\sum_{i}\partial^{2}/\partial x_{i}{}^{2}$ denotes the Laplacian operator.

Then, they demonstrated that by numerically implementing this method, finite element analysis of the crack growth of a blunt notch specimen would not suffer from the mesh sensitivity reported when using local damage modeling. The mesh sensitivity of the local model is a result of the damage process, which tends to localize in a shrinking volume. The singularity of the damage rate at the crack tip, which is generated by the strain singularity at the tip, causes this problematic localization. The material in front of the crack fails instantaneously and in a vanishing volume due to the singular damage rate. Damage rate singularities are eliminated by the nonlocality, which is introduced by the gradient enhancement [27]. The intrinsic length scale generated by the gradient enhancement controls the volume affected by the damage and the crack width. This internal length and the microstructural scale of the material are related, but the exact relationships between them have yet to be fully understood. This also applies to the relationships that regulate damage development, specifically the evolution law and the equivalent strain definition. However, because these relationship must be linked to the microstructural processes that result in damage and fracture, the obtained macroscopic models are predicted to be more accurate than the classical criteria, which mainly ignore these processes.

In 2001, Peerlings et al. [28] compared the effect of adding two different gradient terms to the damage model. Higher-order deformation gradients enter the equilibrium equations directly in one of the gradient terms, whereas the gradient impact follows more implicitly from an additional partial differential equation. While the implicit gradient model quantitatively follows the nonlocal model, the explicit model has a completely different and even nonphysical response in various respects. Then, Peerlings and his colleagues [29] investigated localization issues in local and nonlocal continuum fracture approaches. They argued that applying strong nonlocality to the model would eliminate localization instabilities and damage rate singularities. Consequently, premature crack initiation was minimized and crack growth rates were managed effectively. Weak nonlocality, as given by explicit gradient models, is insufficient for this application.

A symmetric nonlocal damage theory introduced by Borino et al. [30] demonstrated its capacity to generate physically relevant solutions and fully mesh independent outcomes in all circumstances. To fulfill this aim, a new weight function was considered, which is given below.
(10)$$w\left(d,\xi \right)=\left(1-\frac{V_{r}}{V^{*}}\right)\delta \left(d,\xi \right)+\frac{1}{V^{*}}W\left(d,\xi \right)$$
where δ(d,ξ) denoted Dirac delta function.

The difference between this technique and the classic one only becomes obvious near the boundaries; however, far from the boundaries or in the case of unbounded media, the local contribution disappears and this formulation becomes fully equal to the traditional one.

After that, in 2004, Bodin et al. [31] exploited the nonlocal damage model to analyze the fatigue behavior of asphalt concrete. They hypothesized that the amount of tensile stress experienced by the material during mechanical loading would influence the damage progression. The major source of local decohesion in asphalt concrete is believed to be repeated tensile stress. The equivalent strain $\tilde{\varepsilon }\left(s\right)$ can be defined as
(11)$$\tilde{\varepsilon }\left(s\right)=\sqrt{\overset{3}{\underset{i=1}{\sum }}{\left(\frac{\langle \sigma_{i}\rangle }{E\left(1-D\right)}\right)}^{2}}$$
where 〈〉 is the Macauley bracket and $\sigma_{i\left(i=1,\;2,3\right)}$ are principal stresses. Furthermore, the weight function in this case was considered to be:(12)$$W\left(d,\xi \right)=exp\left(-\frac{4d^{2}}{\xi^{2}}\right)$$

The internal length ξ was included to ensure the validity of the constitutive relations during damage localization, which may still occur whenever the loading amplitude is too high, for example, in the case of notched specimens. The internal length influences the size of the damage localization region; the greater this parameter, the longer the fatigue life of bending beams. Additionally, in 2006, Bodin et al. [32] experimentally analyzed the size effect of an asphalt mixture in regard to fatigue evaluation. Substantial agreement was identified between the experimental results and the nonlocal damage fatigue prediction trends. This agreement verified the nonlocal consideration, which allows the effect of a material’s heterogeneous aspect on fatigue failure to be modeled.

Cesar de Sa et al. [33] used a nonlocal gradient damage framework derived from a modified Lemaitre damage model to discuss the characterization of internal material damage in metal-forming procedures like forging or to describe processes wherein fracturing is an independent aspect of the procedure, such as in sheet-blanking or metal-cutting. The damage model accommodates the impact of partial crack closure by changing the damage growth differently under tension or compression. To prevent the mesh size and orientation dependency of the model, a nonlocal damage field is provided that employs a diffusion differential equation, which depends on a characteristic length parameter, as follows:(13)$$\bar{D}-\xi^{2}{▽}^{2}\bar{D}=D$$
where $\bar{D}$ is the nonlocal damage variable that is implicitly related to the constitutive damage parameter, D. Meanwhile, Ubachs, et al. [34] studied the thermal fatigue of a lead-free solder through nonlocal damage modeling and by exploiting the nonlocal damage variable, given as Equation (13). Figure 2 illustrates a comparison of the mesh size dependency of the local and nonlocal models, which shows the damage values and effective plastic strain as a displacement function. As can be seen, when the local model was utilized, the damage became more and more localized as the number of elements increased, whereas if the nonlocal model was used, this unacceptable effect was nearly completely eliminated.

In 2007, Desmorat et al. [35] worked on a three-dimensional damage model for quasi-brittle materials with anisotropy damage, such as concrete. The thermodynamics approach with a single second-order tensorial damage variable was considered, regardless of the loading intensity and signs. Hydrostatic stress was used to record the quasi-unilateral conditions of micro-crack closure. Following that, an efficient strategy for implementing the damage model in commercial finite element programs was discussed, and numerical examples of structural failures were given. The anisotropic damage model demonstrated that it could be employed in large-scale engineering simulations at a reasonable computational cost.

Tovo and Livieri [36] chose the nonlocal implicit gradient technique in 2007 to work on the singularities at the tips of sharp V-notches. A weighted average of a local stress quantity σ derived under the assumption of linear elastic material behavior proved to be a nonlocal equivalent stress $\bar{\sigma }$.
(14)$$\bar{\sigma }-\xi^{2}{▽}^{2}\bar{\sigma }=\sigma$$

They provided an analytical solution for the nonlocal equivalent stress at the crack tip. Various numerical methods can be employed for open notches. They assumed that the material followed a linear elastic constitutive law for welded joints. For assessments of joint fatigue in this context, the nonlocal equivalent stress from the implicit gradient approach was taken into consideration. Both the reliability of fatigue life prediction and the capability to solve complex problems of fatigue life assessment using an “automatic” numerical technique are advantages of the implicit gradient approach over the previous works. 

An integral equation approach based on Equations (4), (6), and (7) was developed by Mallardo [37] to describe elastic-damaging materials. To investigate nonlinear structural problems, including the localization phenomenon, an isotropic damage model was utilized. This approach may be particularly advisable in circumstances where stress or strain concentrations are being observed. Additionally, the method may accurately depict high gradients of stress or strain. The variable chosen for nonlocal consideration was the strain energy release rate. The advantage of this decision was that, notwithstanding the model’s nonlocality, all the constitutive computations could be performed locally at each point x.

A prediction model for evaluating the ductile crack initiation of steel structures was introduced by Kang and Ge in 2013 [38]. A nonlocal damage parameter was presented, based on averaging the strains over the effective plane, as mentioned in Equation (6), and using a weight function in the exponential form. Three different mesh sizes were then applied in finite element analysis. This nonlocal prediction method estimated the ductile crack initiation of steel bridge piers with high precision, irrespective of the specimen structures and loading histories, by comparing the local and nonlocal solutions to the experimental ones. Furthermore, the mesh independent nature of this nonlocal framework was illustrated. Figure 3 depicts a comparison of the local damage model and the proposed nonlocal model. By considering the nonlocal parameter ξ=1, the cumulative damage became more averaged to different degrees.

Lorentz [39] presented a novel nonlocal damage model in 2017, based on constitutive law and the energy dissipation rate given in Equation (2), to characterize the fracture of plain concrete under tensile loading. The nonlocal length scale was conceptualized as the width of the operation region to tackle localized damage and reach the scale of individual cracks. Other parameters such as the tensile strength and fracture energy enabled the model to investigate the initial surface damage and the relationship between damage and stiffness.

In the following year, Nguyen and Nguyen [40] exploited this nonlocal framework to predict the fatigue life of asphalt pavement material. The weight function was considered to be:(15)$$W\left(d,\xi \right)=1-\frac{d^{2}}{\xi^{2}}\;\;if\;d\leq \xi$$

To anticipate the fatigue life of asphalt pavement, the study illustrated the application of an isotropic nonlocal elastic damage model. A damage variable was applied to characterize the damage state of a material at a certain point, and the incorporation of this variable at that point was dependent on both the historical damage state and the current strain tensor. Numerical examples demonstrated that the nonlocal damage theory could be employed to predict the damage evolution of a pavement structure and its service life. Through numerical examples, they also described the potential use of the nonlocal damage theory to predict the damage evolution of a pavement structure and its service life.

In 2020, a nonlocal damage framework was proposed by Feng et al. [41] to analyze quasi-brittle materials stochastically. A list of unknown parameters, such as the material properties and loading behavior, was assimilated and analyzed in diverse operating situations throughout a unified safety assessment framework. A newly created machine learning technique was provided to investigate probabilistic damage analysis problems. In this case, the weight function was considered to be:(16)$$W\left(d,\xi \right)=exp\left(-\frac{d^{2}}{2\xi^{2}}\right)$$

Meanwhile, Livieri and Tovo [42] provided an overview of the geometrical effect on the fatigue strength of steel welded joints by considering nonlocal approaches such as the nonlocal implicit gradient approach. Then, in 2021 [43], they exploited the implicit gradient approach, as mentioned in Equation (14), to study aluminum welded joints numerically. They also discussed various characteristics of aluminum joints and proposed a general fatigue distribution zone for arc-welded joints.

Titscher and Unger [44] demonstrated a gradient-enhanced damage model to simulate high-cycle fatigue. History variables, $\eta \left(\bar{\varepsilon }\right)$, were defined to describe the evolving state of a material. In the case of high cycle fatigue loading, the change of the history variables was negligible in a single cycle, which is known as micro-chronological change. However, this changes significantly (macro-chronological change) over the material’s lifespan and leads to material failure. This model was presented as follows:(17)$$\nabla \sigma =\nabla \left[\left(1-D\left(\eta \left(\bar{\varepsilon }\right)\right)\right)C:\varepsilon \right]=0$$
where ∇ represents the gradient operator. The governing equation of the nonlocal and equivalent strain can be calculated using Equation (9), while the damage variable is a function of η. Figure 4 depicts the evolution of η over the state of strain cycles. In this case, $\eta_{static}$ represents the static condition in which $\dot{\eta }=0$ and $\eta_{fatigue}$ corresponds to this fatigue model.

This model handles damage-induced stress distribution, and the given fatigue parameters allow calibration with the experimental Wöhler lines of a cylindrical specimen.

In 2020, Mareau [45] proposed a nonlocal damage model for metallic material fatigue behavior. This utilized nonlocal damage mechanics to perform a series of constitutive relations, making it applicable to explain both nucleation and initiation. It also explicitly incorporated the progressive stiffness reduction owing to the fatigue crack growth. The performance of the proposed model to predict the effect of loading conditions and post defects on the fatigue behavior of metallic polycrystals was illustrated numerically by implementing this model as a spectrum solver.

In 2020, Pandey et al. [46] introduced and improved the continuum damage model based on the weight function given in Equation (8) and applied it to an extended finite element model to capture the constraint effect during fatigue crack propagation. Their analysis showed that the triaxiality parameter was key to estimating the crack tip constraints correctly. 

In 2021, Kamei and Khan [47] reviewed the use of the nonlocal damage model based on Equations (6)–(8) for the vibrational fatigue and fracture of a structure. 

Reiner [48] used a practical approach for the nonlocal simulation of progressive damage in quasi-isotropic fiber-reinforced composite laminates. In this approach, he applied nonlocal averaging to the damage model as follows:(18)$$\bar{\varepsilon }\left(x\right)=\frac{1}{\Gamma \left(x\right)}\int_{V}W\left(d,\xi \right)\tilde{\varepsilon }\left(s\right)dV\approx \frac{1}{\Gamma \left(x\right)}\overset{n}{\underset{i=1}{\sum }}{\tilde{\varepsilon }}_{i}W_{i}V_{i}$$
where n is the number of discrete values of the nonlocal variable. It was demonstrated that nonlocal modeling allowed for the simulation of more accurate and mesh-independent damage in various applications. It was additionally determined that the nonlocal average radius should be sufficiently small to avoid a considerable increase in computing cost, compared to when local simulation strategies were used.

More recently, Soni et al. [49] examined the fracture behavior of cortical bone, which is significantly affected by its hierarchical structure and a high degree of material anisotropy. A nonlocal gradient-enhanced damage model, as mentioned in Equation (9), was implemented in an iso-geometric setting to predict the fracture behavior of cortical bone.

Meanwhile, Huang et al. [50] investigated and assessed the mechanical characteristics of carbon nanotube-reinforced cement composites using a data-driven machine-learning technique. They used a nonlocal continuum damage model with a weight function similar to Equation (16), and they then employed machine learning to estimate the mechanical characteristics of the nano-material. The experimental data and findings indicated that the machine learning models outperformed the classic response surface methodologies in terms of generalization and predictive effectiveness.

### 2.2. Stress Field Intensity Factor

In 1993, Weixing [51] presented the stress field intensity (SFI) method for assessing fatigue life. This technique used the SFI across the local region of damage, rather than the stress peak value employed in the local stress-strain approach, to quantify the fatigue strength of structures from the macro-mechanics perspective, based on the metal fatigue damage process. Figure 5 depicts the fundamental concept of the SFI method, while the SFI function, $\sigma_{FI}$, was defined as follows:(19)$$\sigma_{FI}=\frac{1}{V}\int_{\Omega }f\left(\sigma \right)w\left(r\right)dV$$
where Ω is the fatigue failure region and V is the volume of Ω. f(σ) is the equivalent stress function, which varies for different materials. 

For elastoplastic materials like carbon steel, aluminum alloy, and titanium alloy, Von Mises equivalent stress, $\sigma_{eq}$, is employed, whereas maximum major stress, $\sigma_{max}$, is used for cast iron and cast steel. Additionally, the weight function 0≤w(r)≤1 physically denotes the contribution of the stresses at point P to the peak stress at r=0, resulting in w(0)=1.

Following that, Qylafku et al. [52] presented an innovative macro-mechanical approach to determine fatigue life while accommodating the evolution of the stress gradient and the elastic-plastic stress distribution. This new model focused on the impact of the principal key aspects affecting the fatigue damage process and was based primarily on the SFI approach. The updated formulation was given as
(20)$$\sigma_{FI}=\frac{1}{x_{ef}}\int_{0}^{x_{ef}}\sigma_{eq}w\left(r\right)dr,\;\;\;\;\;\;\;\;\;\;\;w\left(r\right)=1-\frac{1}{\sigma_{eq}}\frac{d\sigma_{eq}}{dr}r$$

To verify the predictability of this method for fatigue life assessment, it was compared to test results obtained from the rotating bending of specimens with key-seats. Excellent agreement with the respective test results was revealed.

In 2002, Morel and Palin-Luc [53] experimentally analyzed high-cycle multiaxial fatigue for different materials such as mild steel, high-strength steel, and cast iron. They compared the results with this nonlocal model and proved the efficiency of the model.

Some years later, in 2007, Wormsen et al. [54] presented a nonlocal stress approach for fatigue assessment based on the weakest-link theory. In this nonlocal stress technique, the whole stress field was examined, instead of only the highest local stress. The probabilistic distribution of the fatigue strength data from smooth standard materials was used to determine the probability of fatigue damage in a mechanical component under cyclic loading. It was discovered that the nonlocal stress method was related to the probability of identifying the critical fatigue defect in the most intensively stressed volume of the component.

Then, Chamat et al. [55] evaluated the fatigue reliability of a railway wheel subjected to multiaxial non-proportional fatigue loading. For this purpose, combined tension-torsion loading was considered. Based on a stress intensity model, they proposed that effective shear stress $\tau_{ef}$ was as follows:(21)$$\tau_{ef}=\frac{1}{x_{ef}}\int_{0}^{x_{ef}}\tau_{eq}w\left(r\right)dr,\;\;\;\;\;\;\;\;\;\;\;w\left(r\right)=1-\frac{1}{\tau_{eq}}\frac{d\tau_{eq}}{dr}r$$

The experimental fatigue test results showed that their nonlocal criteria predicted reliable equivalent stresses.

After that, Karolczuk [56] applied this nonlocal stress approach to reduce the non-uniform distribution of shear and normal stresses on the critical plane, compared to the uniform distribution. In the reduction procedure, the effects of shear and normal stress gradients on fatigue life were studied. Fatigue tests on hourglass-shaped specimens exposed to combined proportional or non-proportional bending and torsion validated the suggested nonlocal area method. The confirmation was demonstrated using two multiaxial fatigue failure criteria, as well as the experimental and computed fatigue lifespan. In another work, Karolczuk and colleagues [57] analyzed the fatigue crack initiation from a surface using this method. Figure 6 exhibits an example of the effective shear stress distribution in the fatigue failure region. As can be seen, this region is divided by sub-areas, in which $\Omega^{\left(i\right)}=\Delta r\alpha^{\left(i\right)}r^{\left(i\right)}$. Δr, $\alpha^{\left(i\right)}$, and $r^{\left(i\right)}$ were considered to be the ring width, the flare angle of the ring, and the subsequent radius of the shear crack edge.

Next, in 2011, Karolczuk and Blach [58] applied a nonlocal damage parameter based on Equation (19) to conduct a fatigue life assessment of smooth specimens subjected to variable amplitude bending. Utilizing the exponential weight function, the parameter was calculated by averaging the stresses in the material throughout the critical plane. The fatigue lifetimes of specimens exposed to cyclic plane bending were employed to identify the proposed weight function parameter.

To predict the fatigue life of tensile samples with and without notches, Marmi et al. [59] used a multiaxial fatigue damage model based on Equations (19) and (20) to obtain the stress and strain distributions near the notch root. The method described was validated by predicting the fatigue life of notched and holed tensile specimens. This methodology has a significant benefit over traditional fatigue life prediction methods because it can be applied to any realistic or complex component using the finite element method.

In 2012, Zhang [60] theorized a general nonlocal model by manipulating stress field equations and applying effective stress. The relationship between effective stress and elastic strain was found to be strikingly comparable to the stress-strain relationship recognized in nonlocal continuum mechanics, revealing the nonlocal nature of fatigue.

In the following year, Ferre et al. [61] presented fretting nucleation boundaries for three different cylinders, as well as a multiaxial fatigue criterion, by focusing on the stress gradient effect in regard to the crack nucleation of cylinders. The nonlocal stress fatigue analysis results were then compared to the local approach experimentally. This comparison, shown in Figure 7, was obtained by considering local and nonlocal boundaries of Von Mises plastic values computed over a 30 μm diameter volume. Assuming a local approach in which half the test conditions were located, the plastic boundary was found to be related to a constant value of maximal threshold shearing. On the other hand, by averaging Von Mises stress over the grain size volume using the nonlocal approach, the boundary shifted to larger shear stress values.

In 2015, fatigue assessment of welded steel structures under axial loading was undertaken by Baumgartner et al. [62] using the stress averaging approach as follows:(22)$$\sigma_{ef}=\frac{1}{x_{ef}}\int_{0}^{x_{ef}}\sigma \left(r\right)dr$$
where the weight function was considered to be w(r)=1.

In 2020, He et al. [63] and Liao et al. [64] used a stress-based approach, as mentioned in Equation (22), to conduct a probabilistic fatigue assessment of notched components. Experimental data were utilized for model validation and comparison. Although both these studies investigated the influence of the size effect, He et al. [63] theorized a critical distance method and characterized the influence of the size effect on critical distance values $l_{0}=x_{ef}/2$ and performance prediction. Table 1 shows the critical distance calculated for a variety of notched components. Two types of material, namely En3B and Al2024-T351, were tested for different numbers of cycles and stress amplitudes $\sigma_{a}$ to obtain the critical distance values. As can be seen, these data indicate the influence of the size effect on the critical distance.

In 2020, Kang and Luo [65] and Mei et al. [66] reviewed fatigue life prediction models such as continuum and stress-based approaches, which mentioned welded joint and notched components, respectively.

More recently, in 2022, Zhu et al. [67] considered the statistical and geometrical size effects of notched components to propose a generalized weakest-link model for probabilistic life prediction. They studied the probability of material failure at various places using the weight function of the SFI method described in Equations (19) and (20). The size effect of notch fatigue behavior, in particular, was accurately described. The model was validated and compared using experimental data from GH4169 and TC4 notched specimens, as shown in Figure 8, which were obtained for three different probabilities: 10%, 50%, and 90%. The model was compared to the previous weakest-link model and shortened to MHSV for the TC4 notched specimens.

### 2.3. Peridynamics Model

The peridynamics (PD) theory is an extended nonlocal model of classical solid mechanics [68]. The balance of linear momentum is expressed in the PD theory as an integral equation that is generally applied in the presence of material discontinuities like cracks. In contrast, the governing equations of classical continuum mechanics have spatial derivatives that result in singularities at material discontinuities. This theory is comparable to a continuum version of molecular dynamics. The PD model, as shown in Figure 9, provides direct nonlocal interaction between a point x and all points $x^{\prime }$ in its proximity. The radius r centered at x defines the surrounding area of x, and r is referred to as the horizon H. Here, the vector $b=x^{\prime }-x$ is called the bond.

According to the 2005 study by Silling and Askari [69], the acceleration of each particle at point x in the reference configuration at time t can be determined as follows:(23)$$\rho \ddot{u}\left(x,t\right)=\int_{H}f\left(\beta ,b\right)dV+B\left(x,t\right)$$
where f, B, and $\beta =u(x^{\prime },t)-u(x,t)$ are the pairwise force function, relative displacement, and body force density field, respectively.

In a micro-elastic material, the pairwise force function is derivable from scalar micro-potential as follows:(24)$$f\left(\beta ,b\right)=\frac{\partial U_{m}}{\partial \beta }\left(\beta ,b\right)$$
in which the micro-potential $U_{m}\left(\beta ,b\right)$ is the energy in a single bond and has the dimensions of energy per unit volume squared. The local strain energy density is therefore found as follows:(25)$$U=\frac{1}{2}\int_{H}U_{m}\left(\beta ,b\right)dV$$

In 2011, Littlewood [70] investigated the PD model of continuum mechanics as an acceptable method to describe microstructurally microscopic fatigue crack growth. To fully explain the material behavior at the grain scale, an elastic-viscoplastic constitutive model was designed to be applied in a non-ordinary state-based PD using a normalized deformation gradient. The PD method was illustrated using a basic model consisting of a hard-elastic utilization in a single crystal. The combination of the elastic-viscoplastic material model and PD efficiently helped the plastic deformation and damage accumulation modeling in the region of the particle incorporation.

In 2014, Martowicz et al. [71] published the outcomes of numerical analyses of 2D aluminum plate models experiencing crack propagation by employing the PD method. Various numerical evaluations were carried out to demonstrate the reliability and validity of a nonlocal discrete model of continuum mechanics for NDE and QNDE. Besides the theoretical point of view, the acquired results were found to be significantly important for experimental optimization, specifically, sensor distribution and appropriate precision.

In 2016, Zhang et al. [72] investigated the validity of a PD theory for predicting fatigue failure in homogeneous and composite materials. A set of essential damage variables was developed to increase the efficiency and stability of a previously developed PD model for fatigue failure. They also conducted nonlocal region size convergence reports following a revised compact tension test that resulted in curved fatigue crack distributions. The fatigue lifetimes of the PD model were associated sufficiently well with tests of numerous crack growth rates over various cycle ranges. This framework was applied with no modifications to analyze fatigue crack propagation in a two-phase composite with many crack initiation locations and complicated fatigue crack routes. A comparison of the PD model and experimental test for crack paths with three different horizon sizes is shown in Figure 10. Meanwhile, a general surface-based PD model with one geometric dimension much smaller than the other two was formulated by Chowdhury and his co-workers in the same year [73] for quasi-static fracture propagation in a cylindrical shell and flat plates.

Rokkam et al. [74] developed a PD approach to estimate corrosion damage and the ensuing crack propagation processes under the combined influences of corrosion and mechanical loading. The method relies on nonlocal PD theory, which substitutes the governing equations of classical continuum mechanics with integro-differential equations that are easier to solve across singularities such as cracks. For this purpose, a corrosion degradation field Υ(x,t) was used, as in the equation below.
(26)$$\frac{\partial \Upsilon \left(x,t\right)}{\partial t}=\int_{H}\frac{D_{C}\left(x,x^{\prime }\right)\left(\Upsilon \left(x^{\prime },t\right)-\Upsilon \left(x,t\right)\right)}{{\left\|b\right\|}^{2}}dV+J\left(x,t\right)$$
where $D_{C}\left(x,x^{\prime }\right)$ is the effective diffusivity characteristic of corrosion propagation in a material system and J(x,t) is a body flux representing a source of corrosion pitting damage due to the applied boundary conditions. The model was then reduced to provide a PD framework for mechanical corrosion damage and crack propagation. Following that, numerical simulations were utilized to illustrate the PD model’s proficiency in describing the mechanics of corroding solids. The proposed model effectively represents corrosion pitting, nucleation, and the path of crack propagation under the complicated interactions of corrosion and mechanical loads without requiring the domain to be re-meshed or specific numerical treatments. This method offers a physics-based alternative to traditional ideas and enables crack propagation research in corrosive conditions.

More recently, Nguyen et al. [75] suggested a unique energy-based PD model for fatigue cracking. The model presented a characterization of the cyclic bond energy release rate range, as well as the energy-based PD fatigue equations for crack initiation and crack propagation. Different mixed-mode fatigue damage was also examined for verification, and the PD outcomes were validated against the experimental values. Figure 11 shows the predictability of the PD model for fatigue crack growth, compared to the experimental values, over a number of cycles.

Moreover, Nguyen et al. [76] manipulated this model to analyze the fracture behavior of one-dimensional bars and two-dimensional plates through a machine learning (ML) technique. For this purpose, linear regression was utilized to determine the relationships between the displacement of a material point and the displacements of its family members and the applied forces for the machine learning algorithm. The machine learning model was then supplied for coupling the PD model using a numerical procedure. Finally, for both the 1D and 2D structures, the coupling models ML-PD recorded reasonable agreement with both the PD and FEA outputs. These outputs are shown below in Figure 12a,b for a variation of displacements, u and v, along y=W/2 and x=L/2, respectively, as in the geometry of the plate shown in Figure 12c.

PD is a nonlocal theory in which the equilibrium equation of continuum mechanics is rewritten in an integral form, enabling discontinuities to develop spontaneously from the formulation. This offers the potential to provide a general concrete model. In 2021, Hattori et al. [77] reviewed the developments of the PD theory and its application for reinforced concrete materials.

A data-driven method was recently proposed by Ma and Zhou in 2022 [78] to define the nonlocal influence function of bond-based PD, given as ω(b). The data-driven algorithm associated with the Taylor series expansion and the Adam optimization technique in this suggested method demonstrated great stability and excellent accuracy in the complicated nonlocal influence function regression process. The pairwise force function can be rewritten as follows:(27)$$f\left(\beta ,b\right)=\int_{H}\omega \left(b\right)\beta dV$$

Moreover, the algorithm was evaluated using a fabricated dataset, and the measured values were found to correspond satisfactorily with the new nonlocal influence function, proving the practicality of the algorithm.

It is worth mentioning that Javili et al. [79] reviewed and described this framework extensively in 2019, providing an overview of its key technologies and formulations, in addition to related research in many domains, as well as highlighting some research areas that had not yet been evaluated.

### 2.4. Elastic-Plastic Models

In 2002, Polizzotto [80] modified a nonlocal elasticity model by considering nonlocal elastic-plastic materials, in which their total strain tensor was defined as a summation of elastic and plastic strains
(28)$$\tilde{\varepsilon }=\varepsilon^{e}+\varepsilon^{p}$$
where the nonlocal equivalent strain tensor was obtained using Equation (6) and its relation with the nonlocal stress field was considered to be:(29)$$\bar{\sigma }=C:\bar{\varepsilon }$$

Here, C is the usual fourth-order moduli tensor of homogeneous isotropic elasticity, as follows:(30)$$C_{ijhk}=\lambda \delta_{ij}\delta_{hk}+\mu \left(\delta_{ih}\delta_{jk}+\delta_{ik}\delta_{jh}\right)$$
where λ and μ are Lamé constants.

Then, in 2009, Belnoue et al. [81] worked on local-nonlocal damage model of metals by considering the local-nonlocal equivalent plastic strain to be:(31)$${\bar{\varepsilon }}^{p}=m\varepsilon_{nonlocal}^{p}+\left(1-m\right)\varepsilon_{local}^{p}$$
where m is the so-called nonlocal ratio. Meanwhile, the weight function used in this case was considered to be:(32)$$W\left(d,\xi \right)={\left(1-{\left(\frac{d}{\xi }\right)}^{2}\right)}^{2}\;\;\;\;\;\;if\;\;\;\;\;d\leq \xi$$

In a 2015 study by Boeff and colleagues [82] to formulate the nonlocal damage model for complex microstructures, the equivalent plastic strain was considered to be:(33)$${\bar{\varepsilon }}^{p}=\int_{0}^{T}\dot{p}dt$$
where $\dot{p}$ shows the accumulated plastic strain rate, as given below.
(34)$$\dot{p}=\sqrt{\left(\frac{2}{3}{\dot{\varepsilon }}^{p}:{\dot{\varepsilon }}^{p}\right)}$$
where ${\dot{\varepsilon }}^{p}$ represents the plastic strain rate determined by time-derivation. 

The objective was to incorporate a nonlocal damage model into the spectral solver framework and microstructural-scale research to identify vulnerable regions where damage begins to initiate and propagate.

In 2016, Shen et al. [83] proposed a nonlocal method based on continuum damage mechanics to analyze the effect of the stress gradient on fatigue crack initiation. The accumulated plastic strain from Equation (34) was applied and compared to the experimental data. The proposed nonlocal model performed better than the local model.

Kolwankar et al. [84] presented a uniaxial nonlocal formulation for prismatic steel bars. In this formulation, the nonlocal strain was determined using Equations (6) and (31) by considering a bell-shaped weight function. The nonlocal formulation, which was adopted via a one-dimensional structure for necking and buckling, effectively alleviates the mesh dependency of the local models, meaning the former can properly describe the softening load-deformation response independently of mesh discretization. Furthermore, a comparison with finite element test data showed that the nonlocal formulation effectively captured strains in the localized region.

### 2.5. Energy-Based Nonlocal Models

A fatigue criterion suitable for anticipating both the stress gradient and the load effects is required for the successful design of industrial components with regard to high-cycle multiaxial fatigue. These effects play an important role in the transmission of fatigue data from samples to components. In 2003, Banvillet et al. employed an energy-based strategy to provide a new criterion [85]. Using the strain-work density $W_{S}$ assigned to the material as follows:(35)$$W_{S}=\int_{\varepsilon_{min}}^{\varepsilon_{max}}\sigma d\varepsilon$$
and the concept of volume influencing fatigue crack initiation, the new proposed criterion would be applicable for all constant amplitude loadings. The predictions were compared to the volumetric proposals using the local criterion, revealing that the results were extremely accurate and less dispersed than those obtained using the local methods, specifically for loadings with mean stresses or under non-proportional loadings.

Then, in 2013, a fatigue life assessment method was proposed by Saintier et al. [86] for proportional and non-proportional multiaxial variable amplitude. The fatigue criteria for multiaxial constant amplitude loading were based on Equation (35). The aforementioned energy-based fatigue strength criteria were entirely reformulated and expanded, and they had already undergone two significant modifications. The first was a fatigue criterion for multiaxial variable amplitude loadings, although previous researchers have investigated exclusively constant amplitude loadings. The second was an improvement to a progressive fatigue life assessment method for multiaxial variable amplitude loadings with proportional and non-proportional amplitudes. Regardless of the type of variable amplitude loading (uniaxial or multiaxial), no cycle counting method is required. The method was implemented as a post-processor of a finite element program, while experimental results of specimens obtained from the literature were verified using assessments of the method for constant and variable amplitude multiaxial loadings.

Additionally, Krzyzak and Lagoda [87] presented a volumetric methodology and a nonlocal fatigue computation method in 2014 that incorporated an energy parameter. The proposed model was used for simulation computations. The value of the energy parameter $W_{P}$ is defined as follows:(36)$$W_{P}=\frac{1}{2}\bar{\sigma }\left({\bar{\varepsilon }}^{e}+{\bar{\varepsilon }}^{p}\right)$$
in which the equivalent plastic strain ${\bar{\varepsilon }}^{p}$ can be defined using Equation (33).

Both experimental and computational analysis have been subjected to unnotched (smooth) and notched specimens via two types of loads, tension-compression and cyclic bending. The energy parameter values were computed using the finite element method to conduct an elastic-plastic analysis of the cyclic characteristics of a material. It was also noted that the threshold value was determined by the kind of load, and the volume with which the averaging must be conducted is affected by the load level. Figure 13 illustrates the energy parameter over the number of cycles for 10HNAP steel subjected to smooth and notched specimens. The same figure compares the experimental data to the nonlocal averaging values.

In 2016, Meggiolaro et al. [88] introduced a nonlinear incremental fatigue damage model to eliminate the requirement for recording individual loads and utilizing semi-empirical methods for non-proportional amplitude loadings. This methodology, which is based on elastoplastic work, progressively accumulates multiaxial fatigue damage under actual service loads until it reaches 1.0 or any other critical threshold. For uniaxial histories, the method employs the derivative of the normal stress σ with regard to the damage D, denoted here as the generalized damage modulus $D_{G}$.
(37)$$D_{G}=\frac{d\sigma }{dD}\;\;\to \;\;D=\int dD=\int \frac{1}{D_{G}}d\sigma$$

Both the proposed stress and strain-based techniques can be developed using classic stress, strain, or even energy-based damage models, making this strategy an appealing and practical engineering tool. The experimental findings revealed that the suggested strategy can predict multiaxial fatigue life extremely well under complicated tension-torsion histories. Non-proportional tension-torsion tests on tubular 316L stainless steel specimens were used to validate the results.

In the following year, Maurel et al. [89] assessed fatigue crack growth under large-scale yielding conditions. An ideal characteristic length ξ was employed in nonlocal modeling of the strain energy involved in the crack growth process as a consequence of the observation of micro-crack patterns. Using a strain energy analysis based on their 2009 work [90], the selected model partitioned the driving force of crack growth into elasticity, relative to the hydrostatic component of the stress tensor, and plasticity, relative to the shear and deviatoric parts of the stress tensor. The main crack growth rate da/dN formula was given as:(38)$$\begin{array}{c} \frac{1}{\Lambda }\frac{da}{dN}={\left(G_{e}\frac{W_{e}}{\alpha_{e}}\xi \right)}^{m_{e}}+{\left(G_{p}\frac{W_{p}}{\alpha_{p}}\xi \right)}^{m_{p}}\\ W_{e}=\frac{1}{3}\int_{cycle}\langle tr\left(\sigma \right)\rangle .\langle tr\left(d\varepsilon^{e}\right)\rangle \\ W_{p}=\int_{cycle}\tau :d\varepsilon^{e} \end{array}$$
where Λ, $\alpha_{e}$, $\alpha_{p}$, $m_{e}$, and $m_{p}$ are the model parameters. $G_{e}$ and $G_{p}$ are geometrical parameters. $W_{e}$, $W_{p}$, and tr( ) are affiliated to the elastic opening energy, the plastic distortion energy, and the first invariant of the considered tensor, respectively.

In 2019, Raphael et al. [91] evaluated fatigue assessment of short fiber-reinforced thermoplastic materials via combined strain-rate and energy-based fatigue criteria. Based on a vast experimental database, this study examined the evolution of numerous mechanical parameters under cyclic loads. The supplied total strain energy density per cycle $W_{g}$, which was obtained experimentally as the area under the apparent modulus, is referred to as follows:(39)$$W_{g}=\int_{cycle}\langle \sigma \rangle \dot{\varepsilon }dt$$

This criterion yielded particularly acceptable results when compared to the experimental data.

More recently, in 2022, Maurel et al. [92] examined fatigue crack growth under large-scale yielding parameters, exploiting Equation (38) to assess a macroscopic cracked surface based on an explicit depiction of crack growth to evaluate crack-driving forces.

### 2.6. Nonlocal Multiscale Model

Fish and Yu modified the nonlocal damage model in 2002 to conduct a computational analysis of brittle composite materials [93]. The mathematical homogenization theory was developed to account for multiscale damage effects in heterogeneous materials, and a closed-form expression relating nonlocal microphase fields to overall strain and damage was established. Numerical analysis was used to evaluate the accuracy and computational efficiency of the proposed model for both low and high-cycle fatigue. In 2005, the same authors developed a nonlocal multiscale model for fatigue life predictions [94].

Figure 14 demonstrates the macroscopic and microscopic structures of a component. The representative volume elements (RVEs) denoted by Ѳ are assumed to be small compared to the characteristic macro-domain length.

The mathematical homogenization theory with approximate periodic fields is the foundation of this method. The near-periodicity illustrates the accumulating damage caused by permanent deformations in the temporal domain. The initial boundary value problem is divided into micro-chronological (temporal unit cell) $\mu_{t}$ and macro-chronological (homogenized) $\mu_{h}$ problems using various temporal scales. Multiple temporal scales can be defined by the following equation:(40)$$\zeta =\frac{\mu_{h}}{\mu_{t}}$$
where the scaling parameter ζ is defined by the ratio of the characteristic lengths in micro-/macro-chronological time. The flow of plastic strain in this proposed nonlocal model was as follows:(41)$${\dot{\varepsilon }}^{p}=C^{\zeta }\frac{\partial \Phi }{\partial \sigma }$$
where Φ is the rapidly oscillating function that is equal to the response fields, and $C^{\zeta }$ is the consistency parameter formulated by defining the nonlocal consistency parameter via the following well-known integral equation:(42)$${\bar{C}}^{\zeta }\left(x,t\right)=\frac{1}{\Gamma \left(x\right)}\int_{V}W\left(d,\xi \right)C^{\zeta }\left(s,t\right)dV$$

As long as the characteristic size is less than the element size, the nonlocal multiscale model is unresponsive to mesh size. Validation investigations demonstrated some model inadequacies, especially in the modeling of stress-strain loops under high amplitude loading (relatively low cycle fatigue). Figure 15 shows a comparison of multiscale simulations and two different sets of experimental data used for fatigue crack growth prediction over various cyclic loadings. As the figure shows, the model was more compatible with the normal fatigue experiment than the overloading tests performed to investigate the inadequacies of nonlocal multiscale simulation for modeling high amplitude loading in relatively low cycle fatigue. For instance, after almost $7.5\times 10^{4}$ cycles, more than 10% of the errors observed when predicting fatigue crack growth occurred in the overload experiment and multiscale simulation. However, as the number of cycles increased, the nonlocal model showed more compatibility. 

In 2021, Soric et al. [95] reviewed multiscale damage modeling for quasi-brittle and ductile materials by employing nonlocal damage models such as an equivalent strain tensor, as defined in Equation (11). They considered both micro- and macro-levels for finite element discretization, and standard examples were used for verification.

## 3. Extension of Nonlocal Models

This section investigates and reviews other ad hoc models that have occasionally been used in the literature to perform fatigue analyses of solid structures and materials.

A microstructural sensitive model for fatigue analysis was exploited by Owolabi et al. [96] and McDowell et al. [97] to advance fatigue life prediction and structural health monitoring. At the average grain size scale, the fatigue indicator parameters (FIP) gave valuable insights into the microstructure dependency of forces driving fatigue crack growth. The Fatemi-Socie (FS) critical plane parameter, which is such an FIP, is beneficial for assessing fatigue in shear-dominated crack growth and is defined as follows:(43)$$P_{FS}=\frac{\Delta {\bar{\gamma }}_{max}^{p}}{2}\left(1+K\frac{\sigma_{max}^{n}}{\sigma_{y}}\right)$$
where $\Delta {\bar{\gamma }}_{max}^{p}/2$, $\sigma_{max}^{n}$, $\sigma_{y}$, and K are the nonlocal maximum cyclic plastic shear strain averaged over a finite volume of material, the maximum stress normal to the plane of $\Delta {\bar{\gamma }}_{max}^{p}/2$, the cyclic yield strength, and the constant that mediates the influence of normal stress, respectively. Such strategies can be employed to investigate the effects of microstructure characteristics that cause considerable value fatigue responses related to the feathers of probability distributions of prospective surface and subsurface fatigue crack nucleation and growth regions, such as changes in crack formation modes, surface to subsurface transitions, and so forth.

A novel nonlocal lattice particle method (LPM) for three-dimensional elasticity and the fracture simulation of isotropic solids was proposed by Chen and Liu in 2016 [98]. In 2020, Gao et al. [99] investigated this model for the fracture analysis of composite materials. 

Throughout the LPM, a single particle might engage with both neighboring and remote particles, based on the number of layers of particles participating in the interaction distance. For every type of neighbor, the unit cell is determined for a specific interaction distance, and a particle’s potential energy is the total amount of energy associated with these unit cells. At time t, the equation of motion for particle i is given as follows:(44)$$m_{i}{\ddot{u}}_{i}\left(t\right)=\overset{N_{i}}{\underset{j=1}{\sum }}f_{ij}\left(t\right)+b_{i}\left(t\right)$$
where $m_{i}$, u(t), $f_{ij}\left(t\right)$, and $b_{i}\left(t\right)$ are the mass, displacement vector, interaction force between particles i and j, and external force vector, respectively. The interaction force between the particles can be defined as follows:(45)$$f_{ij}\left(t\right)=\frac{\partial U_{i}}{\partial \delta l_{ij}}e_{ij}$$
where $e_{ij}$ and $\delta l_{ij}$ are the unit vector and the distance change between particles, while $U_{i}$ is the stored energy in one of its unit cells, which can be written as follows:(46)$$\begin{array}{c} U_{i}=U_{i}^{local}+U_{i}^{nonlocal}\\ U_{i}^{local}=\frac{1}{2}\overset{n_{i}}{\underset{j=1}{\sum }}k_{ij}{\left(\delta l_{ij}\right)}^{2}\\ U_{i}^{nonlocal}=\frac{1}{2}T_{i}{\left(\overset{n_{i}}{\underset{j=1}{\sum }}\delta l_{ij}\right)}^{2} \end{array}$$
where $k_{ij}$ and $T_{i}$ are local and nonlocal parameter, respectively.

A volumetric nonlocal high cycle fatigue (HCF) strength criterion was proposed by May et al. in 2015 [100] to evaluate the effect of natural corrosion pits on the fatigue of a martensitic stainless steel with high mechanical strength. Nonlocal HCF criteria used the volume averaging technique, which is an approach employed specifically to examine stress-strain distributions for three-dimensional structures. It estimates fatigue life by considering that the observed fatigue strength effect for microscopic cracks is attributable to the fact that the damage processes only occur across a restricted volume. A threshold stress, an energy threshold, or a critical distance threshold can be employed to determine this volume.
(47)$$\left(\bar{\tau },\bar{\sigma }\right)=\frac{1}{V}{∭}_{V}\left(\tau ,\sigma \right)dV$$

This criterion can be represented in the diagram $\left(\bar{\tau },\bar{\sigma }\right)$, which is called the nonlocal Crossland diagram. The investigation concluded that calculating fatigue strength for materials with actual (irregular) pit geometry necessitated a nonlocal HCF strength criterion.

In 2018, a nonlocal approach was proposed based on the optimized effective damage parameter (OEDP) by Li et al. [101] for low cycle fatigue (LCF) life analysis. The effective damage parameter ${\bar{D}}^{ef}$ optimization process was designed to guarantee the finest scatter factor. This nonlocal method was presented with the aim of solving the problem in the weakest-link approach.
(48)$${\bar{D}}^{ef}=(\frac{1}{V}{∭}_{V}D^{b_{W}}dV{)}^{1/b_{W}}$$
where $b_{W}$ is the Weibull shape parameter.

Nonlocal and gradient fracture criteria for brittle, quasi-brittle, and ductile fractures in notched materials were applied by Suknev in 2019 [102]. In this study, a prefecture zone size $d_{f}$ was considered and defined as follows:(49)$$d_{f}=d_{0}+\beta_{p}L_{e}$$
where $d_{0}$ is the representative volume of the material, which is calculated using elasticity theory tools and unrelated to plastic strain, while $L_{e}$ is the stress concentration zone size. $\beta_{p}$ is the plastic parameter: $\beta_{p}=0$ for brittle material, $\beta_{p}\gg 1$ for ductile material, and $\beta_{p}\sim 1$ for moderate plasticity. Therefore, the equivalent stress averaged over the weak section on a scale $d_{f}$ was given as follows:(50)$${\bar{\sigma }}_{eq}=\frac{1}{d_{f}}\int_{0}^{d_{f}}\sigma_{eq}\left(x\right)dx$$

A nonlocal coupled damage-plasticity model was proposed by Nguyen et al. in 2015 [103] via a combination of damage mechanics and plasticity theories to capture the fracture processes of ductile materials. In this framework, the nonlocal equivalent strain rate was defined as follows:(51)$$\bar{\dot{\varepsilon }}\left(x\right)=\left(1-m\right)\dot{p}+\frac{m}{\Gamma \left(x\right)}\int_{V}W\left(d,\xi \right)\dot{p}dV$$

In their model, the Gauss weight function was used, as given below:(52)$$W\left(d,\xi \right)=exp\left(-\frac{\pi d^{2}}{\xi^{2}}\right)$$

The proposed nonlocal coupled damage-plasticity model allowed for pointwise stress updates, thus making it easier to implement existing finite element codes. Numerical examples were also used to show this model’s capability.

A nonlocal fracture criterion introduced by Khodabakhshi et al. [104] in 2016 was developed by the same authors [105] in 2019 and more recently by Alebrahim et al. [106] and Shin et al. [107,108] to investigate the fracture of quasi-brittle materials using a graph-based finite element technique. The main idea was that a network link would fail if the weighted average nonlocal strain, derived using Equation (6), exceeded a critical value, $\varepsilon_{critical}$; i.e.,
(53)$$\bar{\varepsilon }\left(x\right)=e_{i}\cdot \left[\int_{V}W\left(d,\xi \right)\tilde{\varepsilon }\left(s\right)dV\right]e_{i}\geq \varepsilon_{critical}$$
where $e_{i}$ is the unit vector along the edge of each element in the graph-based finite element modeling.

Damage to and fracture of viscoelastic materials were computationally formulated by Thamburaja et al. [109] and Sarah et al. [110] using this criterion. The mesh dependency of the framework and its response to the forced displacement and crack propagation, respectively, were analyzed by these authors.

## 4. Discussion

Despite the significant number of studies already conducted in the field of nonlocal theories for modeling fatigue and fracture of materials and structures, many cases are yet to be investigated. Therefore, discussing a suitable scenario and the existing challenges presented by nonlocal theories would be useful for future studies.

Selecting a nonlocal theory for fatigue and fracture analysis depends on several factors, including the type of material or structure, as well as the parameters considered for investigation. This article assists this decision-making by classifying and introducing various nonlocal theories, giving the parameters and formulation of each theory, and offering comparisons using numerical and experimental approaches. However, because these comparisons were performed independently for each nonlocal model through case studies involving certain materials or structures, it is impossible to provide a quantitative discussion by comparing the nonlocal models.

## 5. Summary

Nonlocal theories such as the nonlocal elasticity theory, nonlocal strain gradient theory, and nonlocal continuum damage model were equipped with intrinsic length scales to be able to analyze size effect phenomena, including fatigue life assessment, crack growth initiation or propagation, and fracture mechanics. Numerous nonlocal theories and models have been introduced in the literature for this purpose, each with its own advantages and disadvantages, as well as its specific application for different materials or structures. 

The current study has attempted to review the nonlocal theories and models that have been introduced and used in the literature regarding the fatigue and fracture criteria of solid structures, namely the continuum damage model, stress field intensity model, peridynamics model, elastic-plastic models, energy-based models, multiscale damage model, microstructure sensitive model, lattice particle model, high cycle and low cycle fatigue damage models, nonlocal coupled damage-plasticity model, and the nonlocal fracture criterion. These mentioned frameworks were investigated and briefly introduced, and their benefits and drawbacks were explained. Also examined were their applications for various structures like two-/three-dimensional structures, bars, plates, beams, and micro-/nano-structures. These can be affected by various kinds of discontinuities; cracks; different types of materials (for example, concrete, quasi-brittle, or stainless-steel materials); and different types of loading (like tension, compression, bending, cyclic, or amplitude loading).

Table 2 provides specific information about the nonlocal models discussed here, such as references, parameters, applications, materials, and structures. 

## Figures and Tables

**Figure 1 materials-16-00831-f001:**
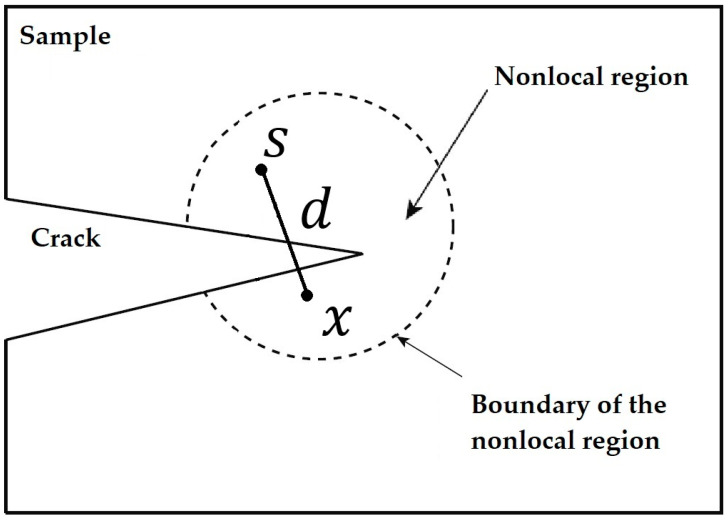
The schematic of a nonlocal region near a crack.

**Figure 2 materials-16-00831-f002:**
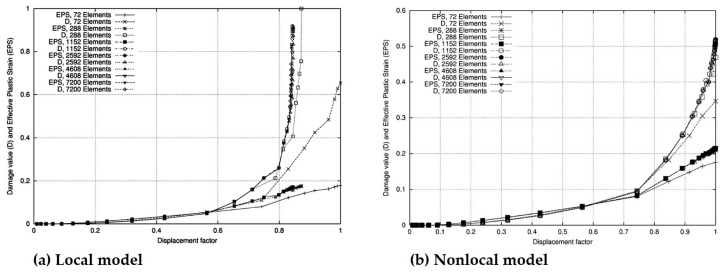
Effect of local and nonlocal models on mesh size dependency [33].

**Figure 3 materials-16-00831-f003:**
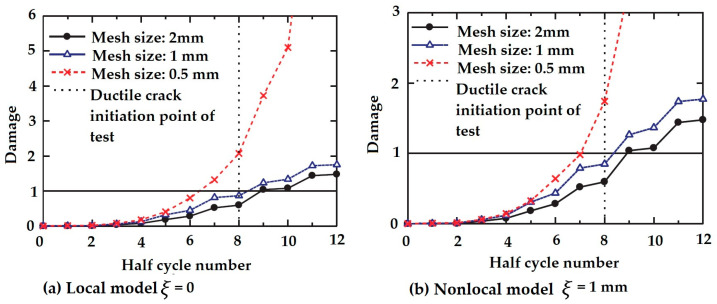
Comparison of local and nonlocal models on damage behaviour [38].

**Figure 4 materials-16-00831-f004:**
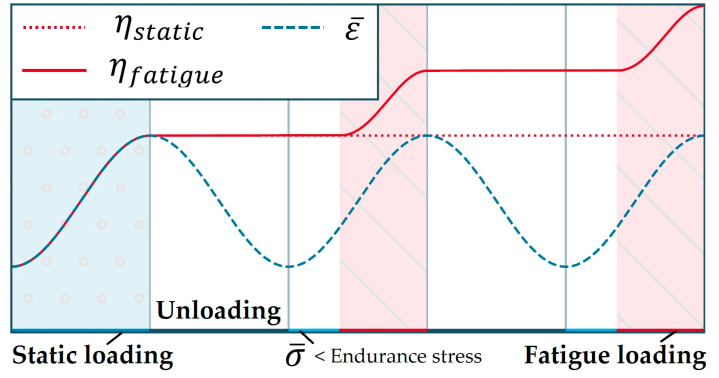
Evolution of history variable η over the state of strain cycles $\bar{\varepsilon }$ [44].

**Figure 5 materials-16-00831-f005:**
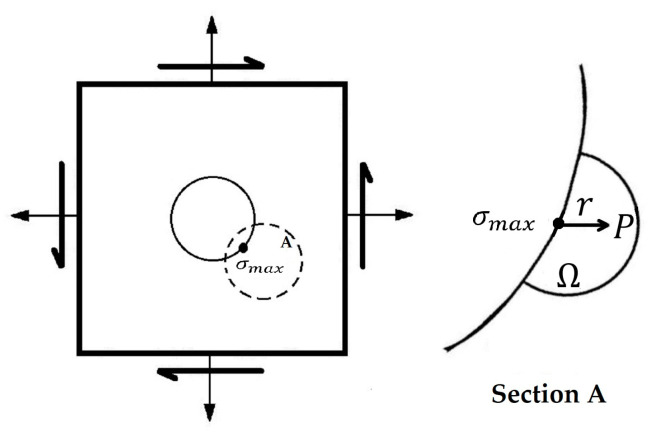
The diagram of the stress field intensity model.

**Figure 6 materials-16-00831-f006:**
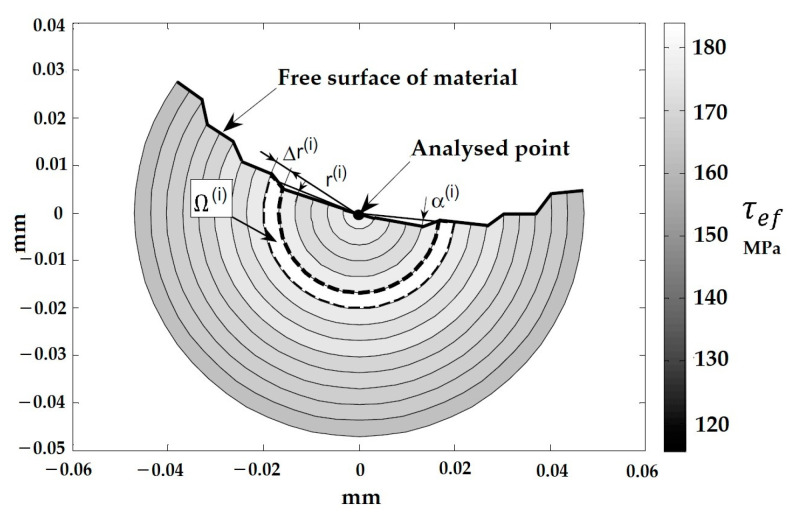
Example of effective shear stress distribution in SFI model [57].

**Figure 7 materials-16-00831-f007:**
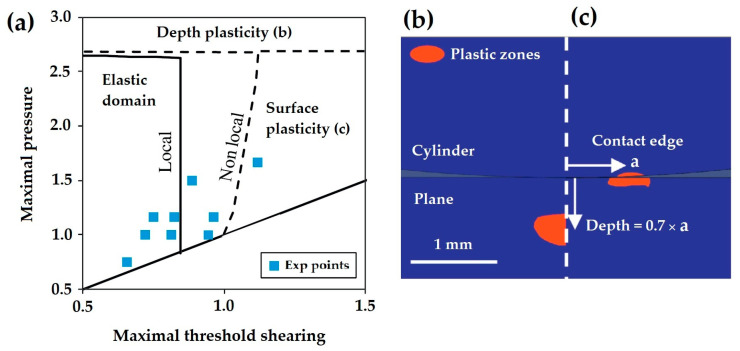
(**a**) Von Mises plastic boundary, (**b**) Depth and (**c**) Surface plasticity zones [61].

**Figure 8 materials-16-00831-f008:**
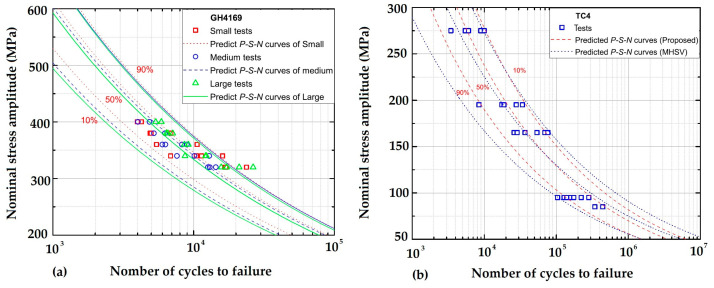
Predicted P-S-N curves for (**a**) GH4169, (**b**) TC4 notched specimens [67].

**Figure 9 materials-16-00831-f009:**
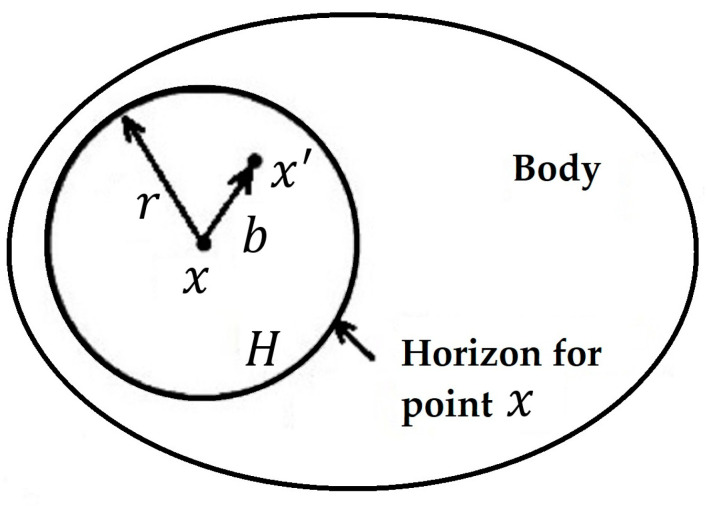
Peridynamics model.

**Figure 10 materials-16-00831-f010:**
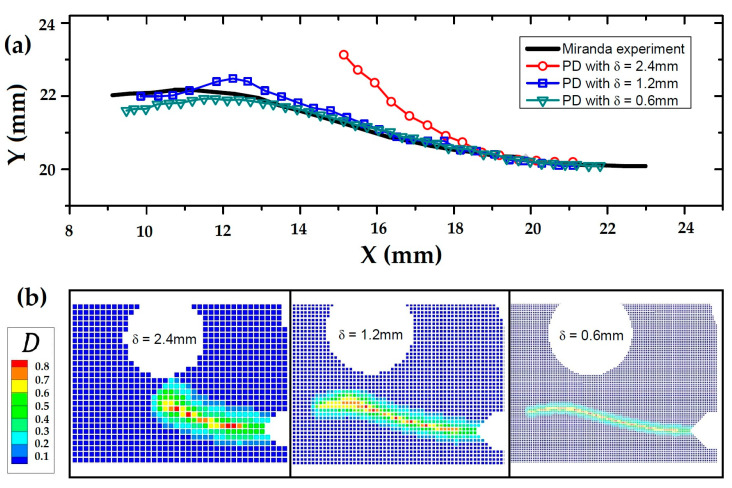
(**a**) PD model and experimental test data for the crack path. (**b**) Different horizon sizes [72].

**Figure 11 materials-16-00831-f011:**
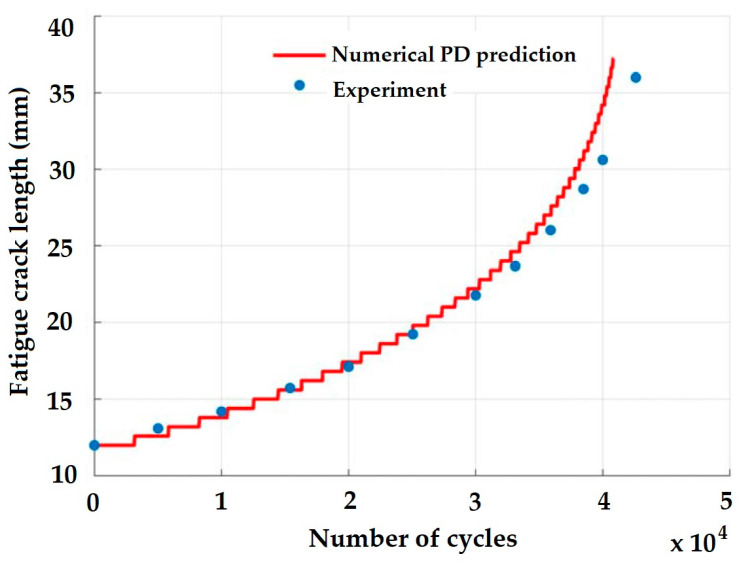
Comparison of PD and experimental data for fatigue crack growth prediction [75].

**Figure 12 materials-16-00831-f012:**
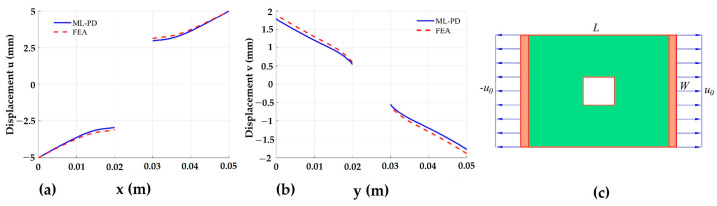
Variation of displacement along (**a**) y and (**b**) x. (**c**) Geometry of plate [76].

**Figure 13 materials-16-00831-f013:**
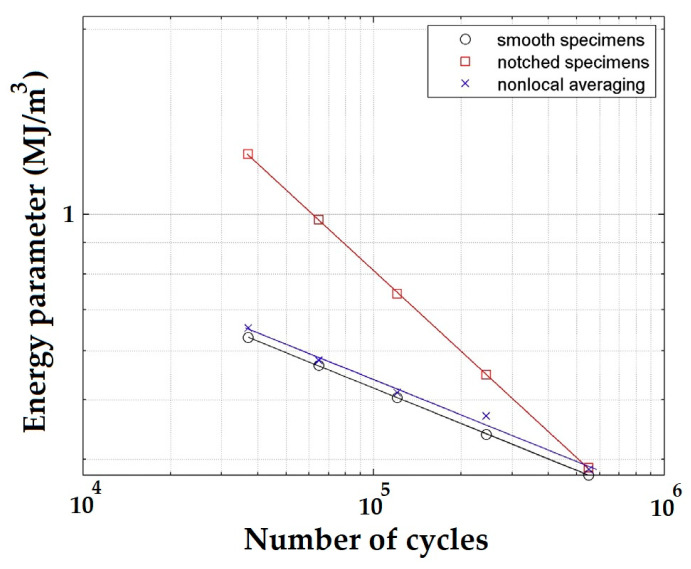
Energy parameter evaluation of 10HNAP steel [87].

**Figure 14 materials-16-00831-f014:**
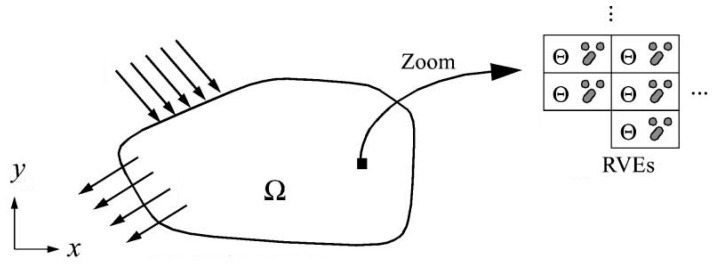
Macroscopic and microscopic structures of a component [93].

**Figure 15 materials-16-00831-f015:**
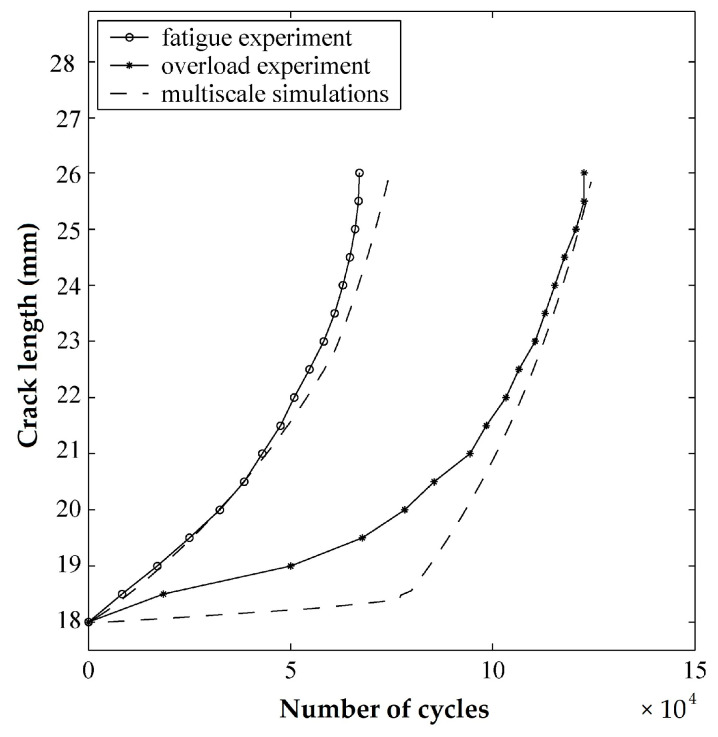
Comparison of multiscale simulations and experimental data [94].

**Table 1 materials-16-00831-t001:** Calculated critical distance for notched specimens [63].

Number of cycles			1 × 10^4^	5 × 10^4^	1 × 10^5^	5 × 10^5^	1 × 10^6^
En3B	V-notched	$\sigma_{a}$ (MPa)	200	131	107	66	52
		$l_{0}$ (mm)	3.315	1.785	1.350	0.720	0.566
	Hole 4 mm	$\sigma_{a}$ (MPa)	190	147	133	101	84
		$l_{0}$ (mm)	3.360	2.740	2.560	2.020	1.500
	Hole 1.75 mm	$\sigma_{a}$ (MPa)	251	195	172	130	106
		$l_{0}$ (mm)	2.640	2.060	1.815	1.420	1.030
Al2024-T351	Radius 1.5 mm	$\sigma_{a}$ (MPa)	157	118	104	79	71
		$l_{0}$ (mm)	0.750	0.588	0.526	0.446	0.456
	Radius 0.25 mm	$\sigma_{a}$ (MPa)	211	161	143	101	97
		$l_{0}$ (mm)	0.307	0.236	0.219	0.206	0.201
	Radius 0.12 mm	$\sigma_{a}$ (MPa)	225	171	151	114	101
		$l_{0}$ (mm)	0.182	0.156	0.110	0.106	0.103

**Table 2 materials-16-00831-t002:** Nonlocal theories in fatigue assessment of solids.

Nonlocal Theories	Applications	Parameters	Materials and Structures	Ref.
NCDM	Damage analysis, Crack analysis, and Life prediction.	$\bar{Y}$ $\bar{D}$ ξ $\bar{\sigma }$ $\bar{\varepsilon }$ η	Elastic bar, Asphalt concrete, Sheet metals, Composites, and Laminates.	[22,23,24,25,26,27,28,29,30,31,32,33,34,35,36,37,38,39,40,41,42,43,44,45,46,47,48,49,50]
SFI	Fatigue life assessment, Size effect.	$\sigma_{eq}$ $x_{ef}$ $\tau_{ef}$	Elastoplastic materials, Steel, Cast iron.	[51,52,53,54,55,56,57,58,59,60,61,62,63,64,65,66,67]
PD	Crack analysis, Plastic deformation, Corrosion.	β Υ(x,t) ω(b)	Elastic-viscoelastic materials, 2D plates, Composites.	[68,69,70,71,72,73,74,75,76,77,78,79]
Elastic-Plastic	Damage analysis, Crack analysis.	m ${\bar{\varepsilon }}^{p}$ $\dot{p}$	Prismatic steel bars	[80,81,82,83,84]
Energy-based	Multiaxial and high cycle fatigue, Crack analysis.	*Λ* $\alpha_{e}$ $\alpha_{p}$ $m_{e}$ $m_{p}$ $G_{e}$ $G_{p}$ $D_{G}$ $W_{P}$	Industrial components, Steel.	[85,86,87,88,89,90,91,92]
Multiscale model	Damage analysis, Crack analysis, Life prediction.	ζ$C^{\zeta }$ Φ	Heterogeneous materials, Quasi-brittle materials.	[93,94,95]
Microstructural sensitive model	Life prediction.	$P_{FS}$	Microstructures.	[96,97]
LPM	Fracture analysis.	$T_{i}$ $U_{i}$	Composite materials.	[98,99]
Volumetric HCF	Corrosion.	$\bar{\tau },\bar{\sigma }$	Martensitic stainless steels.	[100]
OEDP	Low cycle fatigue.	${\bar{D}}^{ef}$	Industrial components.	[101]
Nonlocal and gradient fracture criteria	Fracture analysis.	$\beta_{p}$ $d_{f}$	Brittle, Quasi-brittle, Ductile materials.	[102]
coupled damage-plasticity model	Fracture analysis.	m ${\bar{\varepsilon }}^{p}$ $\dot{p}$	Ductile materials.	[103]
Nonlocal fracture criterion	Fracture analysis, Crack analysis, Computer Simulations.	$\varepsilon_{critical}$	Quasi-brittle materials, Viscoelastic materials.	[104,105,106,107,108,109,110]

## Data Availability

Data is unavailable.

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
