# Peer review of "A Review on Nonlocal Theories in Fatigue Assessment of Solids"

_materials, 2023, doi:10.3390/ma16020831_

Round 1
Reviewer 1 Report
Dear Authors,
The paper can be accepted afer few important improvements.
Specific comments:
a/ not all symbols are explained - please see Eq. 2-3. What is psi? What is phi? angles?
b/Section 2.5 is very limited. Please prepare extensive literature review of energy models. There is a wide group of models.
c/ Section 3 - title is not good "occasionally". Based on this its better to introduce the multiaxiality problem or discuss a possibility of extension non-local models to multiaxial state.
Author Response
The authors are thankful for your valuable comments.
a/ Regarding the explanation symbols in Eq. 2, a phi symbol “ φ “, added to the text before this equation to acquit the misunderstanding. Psi “ ψ “, also mentioned in the text before to be the free energy density.
b/ Some other recent advances and works added to the section 2.5 for a better investigation regarding to the energy-based models.
c/ Based on your comment, the title of section 3 changed to “Extension of nonlocal models”

Reviewer 2 Report
The authors present a nice review of the recent work of the non-local fatigue models. Since the non-local models is less commonly used in the analysis of fatigue, a discussion about the suitable scenario for these models and the existing challenges of the non-local models would be helpful.
Author Response
The authors really appreciated your attention to the article and your precious comment.
Based on your comment a discussion part added to the paper to discuss about a suitable scenario for these models and the existing challenges of the nonlocal models and a comparison of these models tabulated at the end of summary.

Reviewer 3 Report
In this article, the nonlocal theories of fatigue and fracture are presented. Although the author numerous literatures were listed and the relevant theories are of great significance for fatigue life and crack propagation of different structures, the content and writing of the article have major problems. I have to reject this manuscript. Here are the serious problems:
1. The introduction section lacks a detailed introduction of nonlocal theories. The authors only state that the classical theory of elasticity does not apply to fatigue and fracture. It is necessary to explain in detail the differences between nonlocal theory and classical elastic theory. In addition, the multiple theories proposed in the second paragraph do not match the introduction below, and the logical relationship between the theories should be clearly explained.
2. Few pictures make the review article too tedious. There are only two pictures, which are the brief describtion of the methods in the article. In sections 2.1 and 2.2, the authors refer to the experimental results of many researches, but do not provide any tables or pictures describing the data, which may confuse the reader. A good review article must be well illustrated.
3. The writing level of the article needs to be improved. The authors describe the results of the different methods only qualitatively, and the whole article lacks quantitative descriptions. The datas need to be further described after supplementing the corresponding pictures and tables. Therefore, the article could give readers a better understanding of the effectiveness of the different methods.
Author Response
The authors are very grateful for your valuable opinion, which helps to improve and professionalize the article.
- The inadequacies of the introduction have been solved according to your comments and here are the explanation of the modification point by point:
-
- One paragraph added to the introduction as the initial paragraph to explain the fatigue assessment and its relation to the microscopic investigation and size-effect phenomenon.
- The reason of why classical theory of elasticity is insufficient for the microscopic investigation was added to second paragraph. Therefore, this paragraph explains the need for nonlocal theories.
- The third paragraph illustrates how nonlocal theory of elasticity proposed and how it works. Also, the extra explanation of some less related theories was eliminated.
- Next, the differences between classical continuum theory and nonlocal continuum theory stated in the fourth paragraph.
- In the fifth paragraph, continuum damage mechanics introduced and illustrated appropriately.
- Finally, the current study paragraph gives a brief view of what is going to discuss in this paper.
- Thirteen pictures and two tables added to the paper with their descriptions to describe and illustrate the mentioned nonlocal theories and experimental data in a better and more clear way.
- The writing level of the article have been checked and certified by a native English expert. Some quantitative data was added to the article as figures and tables, and experimental data was quantitatively compared to each other. Although comparisons between local and nonlocal models have been made in the research, there is a lack of data in the literature to quantitatively characterize nonlocal models to each other.

Round 2
Reviewer 1 Report
THe paper is improved and can be considered for publication.
Reviewer 3 Report
The authors fully understood the relevant recommendations and made sufficient revisions. I have no other opinion on this article.